# Unique Properties of Heme Binding of the *Porphyromonas gingivalis* HmuY Hemophore-like Protein Result from the Evolutionary Adaptation of the Protein Structure

**DOI:** 10.3390/molecules27051703

**Published:** 2022-03-05

**Authors:** Joanna Kosno, Klaudia Siemińska, Teresa Olczak

**Affiliations:** Laboratory of Medical Biology, Faculty of Biotechnology, University of Wrocław, 14A F. Joliot-Curie, 50-383 Wrocław, Poland; joanna.kosno6@gmail.com (J.K.); klaudia.sieminska@uwr.edu.pl (K.S.)

**Keywords:** *Porphyromonas gingivalis*, HmuY, hemophore-like protein, heme, iron, periodontitis, evolution

## Abstract

To acquire heme, *Porphyromonas gingivalis* uses a hemophore-like protein (HmuY). HmuY sequesters heme from host hemoproteins or heme-binding proteins produced by cohabiting bacteria, and delivers it to the TonB-dependent outer-membrane receptor (HmuR). Although three-dimensional protein structures of members of the novel HmuY family are overall similar, significant differences exist in their heme-binding pockets. Histidines (H134 and H166) coordinating the heme iron in *P. gingivalis* HmuY are unique and poorly conserved in the majority of its homologs, which utilize methionines. To examine whether changes observed in the evolution of these proteins in the Bacteroidetes phylum might result in improved heme binding ability of HmuY over its homologs, we substituted histidine residues with methionine residues. Compared to the native HmuY, site-directed mutagenesis variants bound Fe(III)heme with lower ability in a similar manner to *Bacteroides vulgatus* Bvu and *Tannerella forsythia* Tfo. However, a mixed histidine-methionine couple in the HmuY was sufficient to bind Fe(II)heme, similarly to *T. forsythia* Tfo, *Prevotella intermedia* PinO and PinA. Double substitution resulted in abolished heme binding. The structure of HmuY heme-binding pocket may have been subjected to evolution, allowing for *P. gingivalis* to gain an advantage in heme acquisition regardless of environmental redox conditions.

## 1. Introduction

The development of periodontal diseases is linked with an environmental shift in members of the oral microbiome, leading to the domination of Gram-negative pathogenic bacteria over early Gram-positive commensal colonizers, with species of aerobic *Streptococcus* being the most abundant [1,2]. Bacterial species isolated from subgingival samples associated with the clinical features of chronic periodontitis are characterized by the presence of increased numbers of anaerobic bacteria classified in the red complex, i.e., *Porphyromonas gingivalis* (formerly *Bacteroides gingivalis*), *Tannerella forsythia* (formerly *Bacteroides forsythus*), and *Treponema denticola* [1,3,4,5,6]. Other bacteria, such as *Prevotella intermedia*, a member of the orange complex, serve as early dental plaque colonizers and bridging species with members of the red complex, mainly aiding in creation of an environment that allows for the colonization of subgingival plaque [4,5]. Among them, *P. gingivalis* is considered to be the main etiologic agent of chronic periodontitis. This pathogen is not only responsible for dysbiosis in the oral cavity, leading to the destruction of tooth supporting tissue, but importantly, it is also engaged in the onset and progression of several systemic diseases, including diabetes, rheumatoid arthritis, atherosclerosis, cardiovascular and respiratory diseases, and Alzheimer’s disease [7,8,9,10,11,12,13,14,15].

*P. gingivalis* is a heme auxotroph that must thereby acquire heme as a source of iron (Fe) and protoporphyrin IX (PPIX) [16]. Our previous research resulted in the extensive biochemical and functional characterization of one of the major *P. gingivalis* heme uptake systems (Hmu), and enabled us to understand the mechanism of heme acquisition with the leading role played by HmuY, the first representative of a novel family of bacterial hemophore-like proteins. HmuY sequesters heme from host hemoproteins or heme-binding proteins produced by cohabitating bacteria and delivers it to the TonB-dependent outer-membrane receptor (HmuR), which transports heme through the outer membrane [16]. Our studies revealed a novel heme acquisition mechanism in which oxyhemoglobin is first oxidized to methemoglobin [17], and heme is released due to relaxation of the Fe(III)heme binding affinity of globin [18]. The generation of methemoglobin involves *P. gingivalis* cysteine protease, arginine-specific gingipain A (RgpA) [17], *P. intermedia* cysteine protease, interpain A (InpA) [19], or *Pseudomonas aeruginosa* pyocyanin [10]. Bacteria are then able to fully proteolyze the more susceptible methemoglobin to release free heme, mostly through *P. gingivalis* cysteine protease and lysine-specific gingipain K (Kgp) activity [20,21]. Periodontitis-associated bacteria, especially *P. gingivalis*, which are better adjusted to both anaerobic and aerobic conditions could also benefit from commensal members of the oral microbiome. Hydrogen peroxide produced by *Streptococcus gordonii* converts oxyhemoglobin into methemoglobin [22], thus increasing heme availability. Moreover, heme bound to surface-exposed or secreted *S. gordonii* glyceraldehyde-3-phosphate dehydrogenase may be utilized by *P. gingivalis*, mainly through the sequestration capacity of HmuY [23].

Our theoretical and experimental analyses identified homologs of *P. gingivalis* HmuY in other bacteria [24,25,26,27,28]. In general, the identity of amino acid sequences of these proteins is rather low. Although three-dimensional protein structures of members of the HmuY family identified in the phylum Bacteroidetes are similar overall, significant differences exist in regions corresponding to the heme-binding pocket [26,27,28,29]. This finding was confirmed by our crystallographic studies revealing a unique protein structure of the *P. gingivalis* HmuY protein in both the apoform (PDB ID: 6EWM) [26] and in the complex with heme (PDB ID: 3H8T) [29]. HmuY structure is similar to *T. forsythia* Tfo (PDB ID: 6EU8) [26], *P. intermedia* PinO (PDB ID: 6R2H) [27], and *Bacteroides vulgatus* Bvu (PDB ID:3U22). So far, neither the protein structure of the second HmuY homolog produced by *P. intermedia* (PinA) [27] nor protein structures of HmuY homologs in complex with heme have been determined.

HmuY family proteins function similarly to classical bacterial hemophores that bind heme with high affinity and deliver it to the TonB-dependent outer membrane receptors [30]. However, in contrast to secreted hemophores, HmuY family proteins are mainly composed of β-structures, are associated with the outer membrane of the bacterial cell and outer membrane vesicles through the lipid anchor, and could subsequently be shed from their surface by limited proteolytic processing [26,27,28,29,31]. Proteins belonging to the HmuY family also use different amino acid residues to bind heme. Histidine residues (H134 and H166; numbering of amino acid residues according to the full-length, unprocessed proteins is used) coordinating the heme iron in *P. gingivalis* HmuY [29,32] are unique and generally poorly conserved in the majority of sequences of HmuY homologs identified so far, which mainly utilize methionine residues [26,27,28].

An indepth understanding of virulence factors involved in shifts in microbiomes consortia and subsequent dysbiosis is crucial for keeping homeostasis, preventing, and treating diseases. Importantly, there is an urgent need for the development of new anti-bacterial strategies that would circumvent bacterial resistance against antibiotics, and HmuY is one of such targets. In order to understand the structure-function relationship of proteins, e.g., for drug or therapy development, the detailed characterization of their properties is required. Therefore, the aim of this study was to analyze whether the replacement of histidine residues coordinating the heme iron in the HmuY protein with methionine residues would affect the heme-binding ability of modified protein variants. Since *P. gingivalis* does not produce siderophores or secreted hemophores that allow for efficient iron or heme acquisition, and does not synthesize PPIX, HmuY is the focus of intense studies because of its high expression levels, surface location, high stability and resistance to proteases, efficient heme binding, and, importantly in terms of the host in vivo environment, heme sequestration from host circulatory heme-carrying and -containing proteins produced by cohabitating bacteria [16].

## 2. Results and Discussion

HmuY is the first and best characterized representative of a novel family of bacterial hemophore-like proteins, distinct from a family of classical secreted hemophores [16,30]. Previously, we performed detailed phylogenetic analyses [24,25,26,27,28], which demonstrated that although the proteins forming this unique family are phylogenetically related and structurally similar, differences in amino acid sequences (Figure 1) could result in different structure of heme-binding pockets (Figure 2). The independent evolution of heme binding resulted in different mechanisms of heme coordination in HmuY (two histidine residues) compared to its homologs characterized so far (two methionine residues). We had demonstrated that substitution of both H134 and H166 residues with alanine residues in HmuY resulted in abolished heme binding ability, whereas single substitutions allowed for partial heme binding [32]. Such an effect might result not only from interactions between the protoporphyrin IX ring and the nonpolar heme cavity [29], but also from heme iron coordination through a single histidine residue.

In contrast to HmuY, M162 and M191 in *P. intermedia* PinA or M150 in *P. intermedia* PinO could be directly involved in heme binding [27]. Interestingly, when the heme iron is coordinated by M150 in PinO, both M76 and M176 may interchangeably participate in heme iron coordination. To coordinate heme iron, two methionine residues could also be used by *T. forsythia* Tfo (M143 and M169) [26,28] and by *B. vulgatus* Bvu (M145 and M172) [28].

To examine whether changes observed in the evolution of heme-binding pockets of hemophore-like proteins belonging to the Bacteroidetes phylum might result in improved heme binding ability of HmuY over its homologs [26,27,28], we substituted in the HmuY protein, either singly or in combination, histidine residues with methionine residues. All site-directed mutagenesis variants were overexpressed in *E. coli* cells and purified similarly to the native HmuY protein (Figure 3a) [26,27]. Far-UV CD spectroscopic analysis of HmuY variants demonstrated that site-directed mutagenesis did not influence their secondary structures (Figure 3a). Theoretical analysis of overlapped three-dimensional protein structures of HmuY variants confirmed that amino acid substitutions did not significantly influence their overall tertiary structures (Figure 3b). However, it is possible that the introduction of methionine residues instead of native histidine residues may cause steric hindrance between M134 and Y127 in apo-HmuY (Figure 3c) and between M134 and M136 in the HmuY-heme complex (Figure 3d). This can cause local distortions, leading to inappropriate heme positioning with respect to binding ligands. In addition, the closer location of M134 and M166 in the HmuY-heme complex as compared to that of H134 and H166 in the native HmuY-heme complex (Figure 3e) can reduce the space inside the heme-binding pocket, which may prevent the heme from entering the pocket.

Analysis of UV-visible absorption spectra revealed significant differences in heme binding to HmuY variants under air (oxidizing) conditions (Figure 4a). Compared to the native HmuY, all HmuY variants constructed in this study bound heme with significantly lower ability as compared to the native HmuY protein, but in a similar manner as compared to *B. vulgatus* Bvu [28], as well as to a lesser extent to *T. forsythia* Tfo [26]. This finding could suggest lower ability to bind Fe(III)heme through the methionine ligand/s under these conditions.

Analysis carried out under reducing conditions demonstrated that single histidine substitution by methionine did not significantly affect heme binding to the HmuY variants (Figure 4b). UV-visible absorption spectra were similar to those obtained for the native HmuY, *T. forsythia* Tfo [26], *P. intermedia* PinO and PinA [27], and *B. vulgatus* Bvu [28], suggesting that a mixed histidine-methionine amino acid couple could be sufficient to at least partly bind Fe(II)heme. Reducing conditions may influence the properties of iron coordination by methionine residues more effectively as compared to histidine residues, demonstrating that methionine-ligand binding would be destabilized under oxidizing conditions [37,38,39]. According to our previous theoretical analysis, other *Porphyromonas* species (e.g., *P. uenonis*, *P. endodontalis*) could bind heme using histidine and another amino acid (e.g., methionine, arginine, leucine, glutamine) [26], which suggests the evolutionary adaptation of the protein structure, leading to more efficient heme binding and heme sequestration by *P. gingivalis* HmuY. The possibility of steric hindrance with tyrosine at position 127 (Y127) due to the substitution of H134 with methionine in the apo-protein cannot also be excluded (Figure 3c). Surprisingly, substitution of two histidine residues by methionine residues resulted in significantly affected heme binding ability, even under reducing conditions. It seems that this replacement appears to have reduced the space inside the heme-binding pocket, preventing heme from entering the pocket. Indeed, when theoretical analysis of tertiary structure of the protein in complex with heme was carried out, steric hindrance between M134 and native M136 can be observed (Figure 3d).

HmuY binds heme effectively under oxidizing conditions with a *K*_d_ value < 10^−9^ M, and this tendency is preserved under reducing conditions (*K*_d_ ~10^−8^ M) [26,27]. Homologs produced by *T. forsythia*, *P. intermedia* and *B. vulgatus* bind heme with significantly lower ability under oxidizing conditions, whereas reduction results in significantly higher heme binding ability, comparable to that observed for the HmuY protein [26,27,28]. Compared to the native HmuY, H134M (*K*_d_ = 1.2 ± 0.3 × 10^−5^ M) and H166M (*K*_d_ = 2.3 ± 0.1 × 10^−7^ M) variants bound heme under oxidizing conditions with significantly lower ability (Figure 5a). However, under reducing conditions, H134M (*K*_d_ = 1.1 ± 0.3 × 10^−8^ M) and H166M (*K*_d_ = 2.2 ± 0.5 × 10^−9^ M) protein variants bound heme with higher efficiency (Figure 5b). H134 may be more important for heme iron coordination than H166 since weaker binding was observed in the case of the H134M protein variant (Figure 4 and Figure 5). However, we did not exclude the possibility of steric hindrance between M134 and Y127 (Figure 3c,d), resulting in lower ability of heme binding by the H134M protein variant. Double histidine substitution either by alanine [32] or by methionine (Figure 4) examined under both redox conditions resulted in abolished heme binding, making proper determination of the heme dissociation constant impossible. We assumed that the introduction of two methionine residues instead of two histidine residues may cause structural changes in this region of the heme-binding pocket (Figure 3e), resulting in reduced space between two introduced methionine residues, thus precluding efficient heme binding.

The PPIX has a planar structure that can accommodate a variety of metals. Metalloporphyrins differ in distance between the center metal atom and the pyrrole nitrogen atom, as well as in-plane or out-of-plane metal location [40]. This results in different modes of metalloporphyrins binding, also in the case of the HmuY protein [41]. Since double HmuY site-directed mutagenesis variant, H134M/H166M, did not bind heme, we examined whether it would bind unsubstituted PPIX ring. As shown in Figure 6a , the H134M/H166M variant incubated with equimolar concentration of PPIX bound this compound with the highest efficiency. To confirm these findings, all proteins were incubated with PPIX, and unbound PPIX was removed by desalting. The H134M/H166M variant bound PPIX with the highest efficiency (Figure 6b), suggesting that the smaller heme-binding pocket caused by double amino acid substitution is able to accommodate PPIX ring.

## 3. Conclusions

*P. gingivalis* HmuY may have been subjected to evolution resulting in acquiring two histidine residues compared to two methionine residues in its homologs characterized so far, resulting in the unique structure and properties of the heme-binding pocket. This feature may allow for *P. gingivalis* to gain an advantage in heme acquisition regardless of environmental redox conditions when the bacterium grows not only in the complex oral microbiome, but also when it transfers and invades other niches of the human body. From a practical point of view, our study might help to develop novel therapeutic strategies on the basis of a Trojan horse strategy to defend anaerobic infections that are currently difficult to treat in the light of increasing antibiotic resistance. Noniron metalloporphyrins mimic heme and can thus be acquired by bacteria using their heme uptake systems [42,43]. In addition, they are good candidates for biomedical or biotechnological applications for therapeutic purposes, especially when they are bound to unique bacterial proteins [44]. HmuY homologs, which use methionine residues to bind heme, did not exhibit ability to bind noniron metalloporphyrins (our unpublished data). HmuY, with its high ability to effectively complex not only heme but also some noniron metalloporphyrins, including Ga(III)PPIX, Co(III)PPIX, Cu(II)PPIX, results in reduced *P. gingivalis* growth and infection ability [41], and it could be used to selectively bind and deliver these compounds as antibacterial agents or photosensitizers to specifically target *P. gingivalis*.

## 4. Materials and Methods

### 4.1. Site-Directed Mutagenesis

Single (H134M or H166M) or double substitutions (H134M/H166M) were introduced into the *hmuY* gene using a Q5 Site-Directed Mutagenesis Kit (NEBaseChanger, New England Biolabs, Ipswich, MA, USA). For this purpose, the expression plasmid possessing the entire *hmuY* gene was used [26] as a template and primers designed in this study. To replace histidine at position 134 with methionine, forward (5’ACCTGATGGTATGAAGAAC3’) and reverse (5’CCCATTTCGTACTTGACTG3’) primers were used. To replace histidine at position 166 with methionine, forward (5’GGAATTCTCTATGGGTCCTGCCGGTCCCACTTAC3’) and reverse (5’AGCCAACCACCTGAAGCA3’) primers were used. The entire procedure was carried out according to the manufacturer’s protocol (New England Biolabs).

### 4.2. Protein Overexpression and Purification

*Escherichia coli* Rosetta (DE3)RIL strain (Agilent Technologies, Santa Clara, CA, USA) was cultured under standard aerobic conditions. HmuY protein or its site-directed mutagenesis variants, lacking the predicted signal peptide sequence, were overexpressed in *E. coli* cells and purified from a soluble fraction obtained from *E. coli* cell lysate as described previously [26]. Concentration of the purified protein variants was determined using the modified Bradford method [45] with RotiNanoquant reagent (Roth, Frederikssund, Denmark). Purity of protein samples was examined using sodium dodecyl sulfate-polyacrylamide gel electrophoresis (SDS-PAGE) and Coomassie Brilliant Blue (CBB) G-250 staining as reported previously [26].

### 4.3. Heme and PPIX Binding Experiments

Heme (hemin chloride; Pol-Aura, Olsztyn, Poland) solutions were prepared as reported previously [26]. The formation of protein-heme complexes was examined in 20 mM sodium phosphate buffer, pH 7.4, containing 140 mM NaCl (PBS) after incubation of proteins with heme for 1, 10, 20, and 30 min. No significant changes in the UV-visible absorption spectra were observed with respect to incubation time. Therefore, the spectra were recorded after 1 min incubation of protein with added heme in the range of 250–700 nm with a double-beam Jasco V-650 spectrophotometer using cuvettes with 10 mm path length. To generate redox conditions, 10 mM sodium dithionite prepared in PBS was used as the reductant, and samples were examined under a mineral oil overlay [27]. Proteins were analyzed at 5 μM concentration.

PPIX (Sigma-Aldrich, St. Louis, MO, USA) stock solution was prepared by dissolving PPIX in pure DMSO (99.9%) (Fluka, Munich, Germany) and subsequently diluting it in PBS. The formation of protein-PPIX complexes was examined in PBS. Alternatively, protein-PPIX complexes were prepared by incubating 150 μM stock solution of proteins in PBS at a 1:1.2 protein:PPIX ratio and subsequently passed through Zeba^TM^ Spin Desalting Columns (Thermo Scientific, Waltham, MA, USA) to unsure that un-complexed PPIX remained. Protein samples were then diluted to obtain 10 μM protein concentration. UV-visible spectra were recorded in the range of 250–700 nm with a double-beam Jasco V-650 spectrophotometer using cuvettes with 10 mm path length.

To determine dissociation constants (*K*_d_), 5 μM protein samples were titrated with heme and UV-visible difference spectra (ΔAbsorbance) between the protein + heme and heme-only samples were recorded. Samples were analyzed under air (oxidizing) conditions (λ = 413 nm) or reduced by sodium dithionite (λ = 424 nm). Titration curves were analyzed using the equation for a 1-site binding model, and *K*_d_ values were determined as reported earlier [26,27] using OriginPro 2020 software (OriginPro Corporation, Northampton, MA, USA). Results are shown as mean ± standard deviation (mean ± SD) from two independent experiments.

### 4.4. Far-UV Circular Dichroism (Far UV CD) Spectroscopy

Proteins were prepared in 10 mM sodium phosphate buffer, pH 7.4, containing 20 mM NaCl. The protein concentration was adjusted to 2.5 μM. Far-UV CD spectra were recorded in the range of 200–260 nm at 25 °C using a Jasco J-715 spectropolarimeter with a scan speed of 100 nm min^−1^, response time of 2 s, and a slit width of 1.0 nm. Mean spectra were calculated from three independently recorded datasets.

### 4.5. Bioinformatic and Statistical Analyses

Amino acid sequences were compared using the Multiple Sequence Alignment Clustal Omega available from the European Bioinformatics Institute (EMBL-EBI) [33]. Three-dimensional protein structures were visualized using the UCFS ChimeraX available from the UCSF Resource for Biocomputing, Visualization, and Informatics (http://rbvi.ucsf.edu, accessed on 1 February 2022) [34]. Three-dimensional models of HmuY site-directed mutagenesis variants were constructed using the Protein Homology/analogY Recognition Engine 2.0 (Phyre2; http://www.sbg.bio.ic.ac.uk/phyre2, accessed on 1 February 2022), and the protein structures were deposited in the RCSB PDB database (PDB IDs: 6EWM and 3H8T) as templates [35]. The model of modified heme-binding pocket was generated using Swiss-PdbViewer (http://www.expasy.org/spdbv, accessed on 1 February 2022), and the protein structures were deposited in the RCSB PDB database (PDB IDs: 6EWM and 3H8T) as templates [36]. Statistical analysis was performed using Student’s *t* test and GraphPad software (GraphPad Prism 5.0 Inc., San Diego, CA, USA).

## Figures and Tables

**Figure 1 molecules-27-01703-f001:**
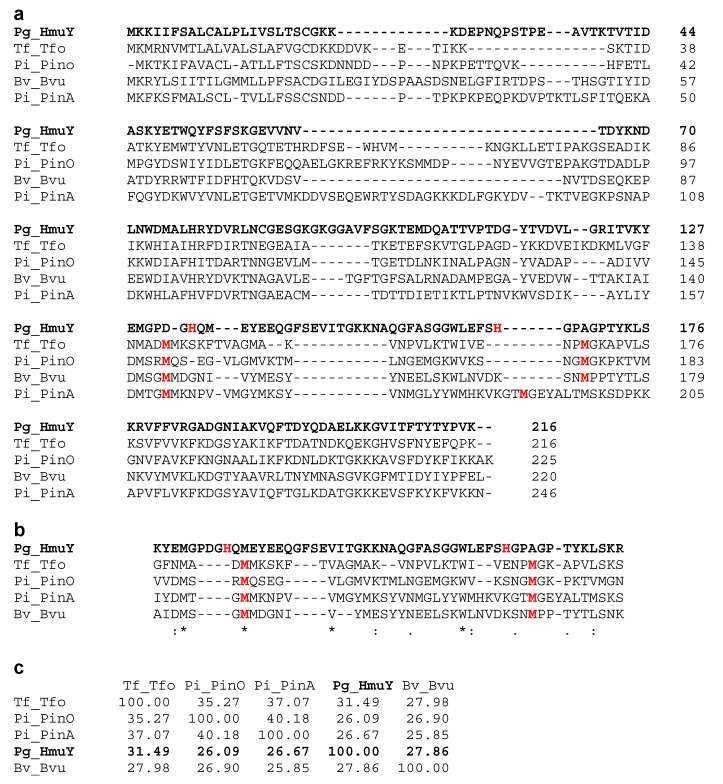
Comparison of best-characterized HmuY family proteins. (**a**) Amino acid sequence alignment of full-length proteins, (**b**) their regions comprising heme-binding site, and (**c**) determination of amino acid sequence identity, carried out using EMBL-EBI Clustal Omega tool [33]. Data obtained for HmuY shown in bold. Amino acid residues coordinating heme iron, identified by site-directed mutagenesis, shown in red. Tf, *T. forsythia*; Pi, *P. intermedia*; Pg, *P. gingivalis*; Bv, *B. vulgatus*.

**Figure 2 molecules-27-01703-f002:**
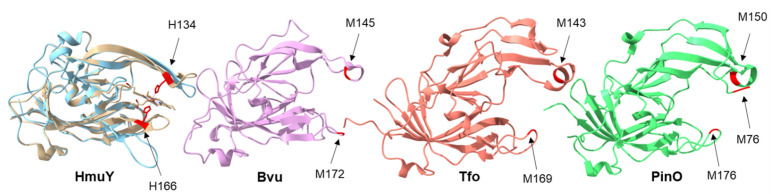
Comparison of three-dimensional protein structures of best-characterized HmuY family proteins. Tertiary structures of proteins determined by X-ray crystallography shown. *P. gingivalis* apo-HmuY (blue) (PDB ID: 6EWM) [26] and HmuY in complex with heme (beige) (PDB ID: 3H8T) [29], *B. vulgatus* Bvu (purple) (PDB ID:3U22), *T. forsythia* Tfo (coral) (PDB ID: 6EU8) [26], and *P. intermedia* PinO (green) (PDB ID: 6R2H) [27]. Amino acid residues coordinating heme iron, identified by site-directed mutagenesis, are shown in red and indicated with arrows. Three-dimensional protein structures visualized using UCFS Chimera tool [34].

**Figure 3 molecules-27-01703-f003:**
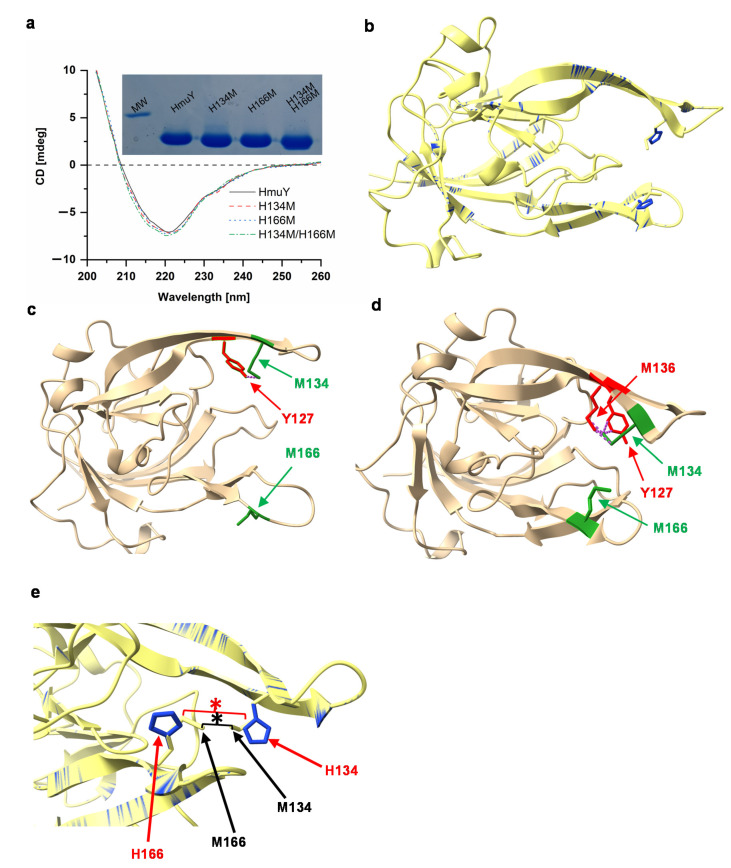
Characterization of HmuY site-directed mutagenesis variants. Proteins were overexpressed, purified, and analyzed by SDS-PAGE and CBB G-250 staining (**a**, inset). (**a**) Secondary protein structures of purified apoproteins determined by far-UV CD spectroscopy and (**b**) tertiary protein structures of the overlapped native HmuY in complex with heme (blue) (PDB ID: 3H8T) and the H134M/H166M protein variant (yellow) analyzed by theoretical modelling [35,36]. (**c**) Modeled three-dimensional apoprotein structure (PDB ID: 6WEM), (**d**) protein structure of HmuY-heme complex (PDB ID: 3H8T), and (**e**) its heme-binding pocket. (**c**) Potential steric hindrance between methionine at position 134 (M134, green) and tyrosine at position 127 (Y127, red) in the apo-HmuY structure (PDB ID: 6WEM). Potential steric hindrance between M134 and M136 in the HmuY-heme complex structure (PDB ID: 3H8T). (**e**) Substitution of two histidines (shown by red arrows) by two methionines (shown by black arrows) may result in reducing the space of the heme-binding pocket. Reduced distance indicated by red and black asterisks.

**Figure 4 molecules-27-01703-f004:**
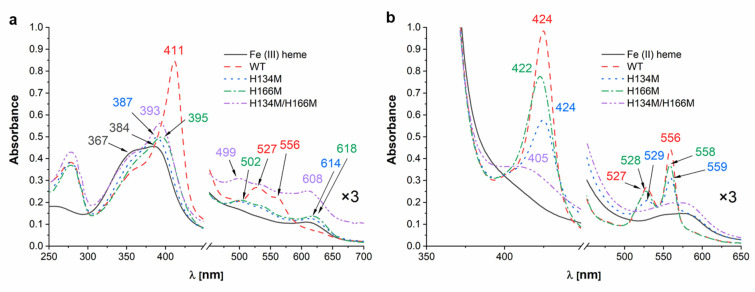
Analysis of heme binding to the site-directed mutagenesis HmuY variants. UV-visible absorption spectra of proteins in complex with heme (1:1 protein:heme ratio) recorded under (**a**) air (oxidizing) conditions or (**b**) reducing conditions formed by addition of sodium dithionite. Absorbance spectra at Q band region magnified three times (×3).

**Figure 5 molecules-27-01703-f005:**
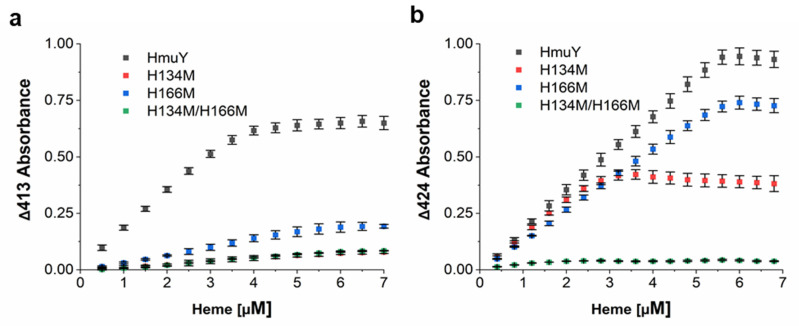
Heme binding ability to site-directed mutagenesis HmuY variants. Curves generated after titration of 5 μM protein samples with heme by measuring UV-visible difference spectra between protein + heme and heme-only samples. Samples analyzed under (**a**) air (oxidizing) conditions or (**b**) reduced by sodium dithionite. Results shown as mean ± SD from two independent experiments.

**Figure 6 molecules-27-01703-f006:**
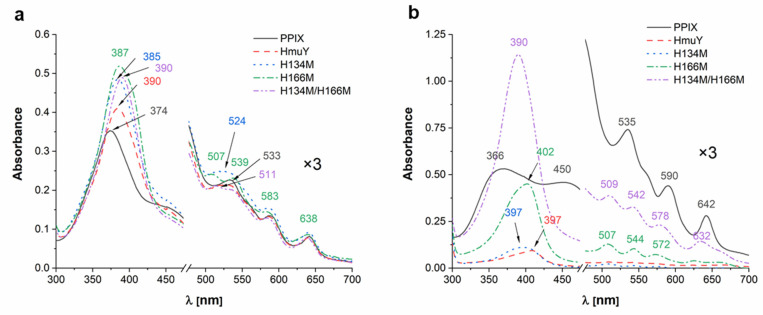
Analysis of PPIX binding to site-directed mutagenesis HmuY variants. UV-visible absorption spectra of proteins in complex with PPIX (1:1 protein:PPIX ratio) were recorded (**a**). Proteins were also incubated with PPIX (1:1.2 protein:PPIX ratio), unbound PPIX was removed by gel filtration, and UV-visible absorption spectra were recorded (**b**). Absorbance spectra at the Q band region are magnified three times (×3).

## Data Availability

Data are contained within the article and are available from the corresponding author upon request.

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
