# Peer review of "Unique Properties of Heme Binding of the Porphyromonas gingivalis HmuY Hemophore-like Protein Result from the Evolutionary Adaptation of the Protein Structure"

_molecules, 2022, doi:10.3390/molecules27051703_

Round 1

Reviewer 1 Report

This manuscript reports unique properties of Porphyromonas gingivalis HmuY, a hemophore-like protein which has been studied for years by the team. The two histidine residues coordinating the heme, locating at the heme-binding pocket P. gingivalis HmuY, are unique and poorly conserved in the majority of its homologs in many other periodontal pathogenic bacteria, in which the function is mediated commonly by methionines. By using the site-directed mutagenesis, authors revealed an advantage in heme acquisition for His (one of those two residues) under both oxidizing and reducing conditions, and also a reduced affinity to PPIX. Authors proposed that the distinct feature in HmuY is probably an evolutionary result of adaption.

The article demonstrates the superiority of two histidines on heme-binding pocket of HmuY in redox environmental adaptation compared with the conserved methionines in its homologs, and provides valuable information for evolution of similar hemophore-like proteins and researches on novel antibiotic.

Overall, the manuscript is high quality and well written. Here are some comments for improments:

  1. It is very difficult to get an estimate how conservative the sequences showed in Figure 1 are. Please add additional information showing sequence conservation, especially for critical regions.
  2. For the experiments operated to exam heme-binding efficiency, if the coordination of heme and protein was weaker under oxidizing condition, then the incubation time of protein and heme should be given. Although the missing peak around 412 nm (Figure 4 left) after site-directed mutagenesis may be due to the residue substitution, as proposed, it may possibly be due to the slower rate for binding.
  3. In Figure 5 (right), the data presented here may also be affected by binding rate. Thus, whether a time for the binding assay is sufficiently long may lead to significantly different results.
  4. An explanation of how the differences on heme-binding efficiency under oxidizing and reducing conditions after residue substitution were caused by steric hindrance is needed.
  5. Line 243, Authors state that the none-iron metalloprophyrins mimic heme might be good candidates of antibiotic for those bacteria using their heme-uptake system to acquire iron. It would be logic that evolution fixes the substitution of Met to His for binding heme with higher specificity, which may exclude the influence of those natural none-iron metalloprophyrins.

Reviewer 2 Report

  Kosno and co-workers present in this work the study and characterization of the mutation of two relevant Histidine residue (H134 and H166) participating in the stabilization of heme group in HmuY from P. Gingivalis. Those Hist residues are poorly conserved in the majority of its homologs while playing an essential role in the binding affinity of the heme group. They mutated the Hist residues to the most abundant Methionine and founded that the single and double mutant tune the ability to bound Fe(III) and Fe(II) heme, and abolished heme binding in the later case.

  They provided sufficiently amount of data to support their conclusion and the article is well written. Still there is room for improvement if presentation is better set. Figures are in general hard to see, i.e. small labels, etc. In particular Figure 3 is somehow chaotic. It has 6 panels but only 3 are label a, b and c. The panel with the dark background should definitively be improved. You may use Chimera or CCP4-MG, etc. Figure 4 is clear but 5 and 6 are not. In Fig.6 have two “x3” ¿what that means?

  Other than that I think this article will be of interest for the community.
